# The Effects of Vitamin D on Immune System and Inflammatory Diseases

**DOI:** 10.3390/biom11111624

**Published:** 2021-11-03

**Authors:** Tomoka Ao, Junichi Kikuta, Masaru Ishii

**Affiliations:** 1Department of Immunology and Cell Biology, Graduate School of Medicine and Frontier Biosciences, Osaka University, Osaka 565-0871, Japan; tomoka_ao1203@icb.med.osaka-u.ac.jp (T.A.); jkikuta@icb.med.osaka-u.ac.jp (J.K.); 2WPI-Immunology Frontier Research Center, Department of Immunology and Cell Biology, Osaka University, Osaka 565-0871, Japan; 3Laboratory of Bioimaging and Drug Discovery, National Institutes of Biomedical Innovation, Health and Nutrition, Osaka 567-0085, Japan

**Keywords:** vitamin D, immune system, COVID-19, rheumatoid arthritis (RA), systemic lupus erythematosus (SLE), multiple sclerosis (MS)

## Abstract

Immune cells, including dendritic cells, macrophages, and T and B cells, express the vitamin D receptor and 1α-hydroxylase. In vitro studies have shown that 1,25-dihydroxyvitamin D, the active form of vitamin D, has an anti-inflammatory effect. Recent epidemiological evidence has indicated a significant association between vitamin D deficiency and an increased incidence, or aggravation, of infectious diseases and inflammatory autoimmune diseases, such as rheumatoid arthritis, systemic lupus erythematosus, and multiple sclerosis. However, the impact of vitamin D on treatment and prevention, particularly in infectious diseases such as the 2019 coronavirus disease (COVID-19), remains controversial. Here, we review recent evidence associated with the relationship between vitamin D and inflammatory diseases and describe the underlying immunomodulatory effect of vitamin D.

## 1. Introduction

Vitamin D deficiency, which causes an imbalance in bone remodeling, is a global public health problem and its frequency is increasing. Due to the pleiotropic effects of vitamin D, its deficiency is related to a higher risk of cardiovascular diseases [1,2,3], infectious diseases, and autoinflammatory diseases, such as rheumatoid arthritis (RA), systemic lupus erythematosus (SLE), and multiple sclerosis (MS). In addition, vitamin D taken for the treatment and prevention of disease has been debated, given its immunosuppressive effect. Anti-cancer effects of vitamin D have found application in cancer treatment [4]. Recent studies have shown that immune cells, such as monocytes, macrophages, dendritic cells, and lymphocytes, express the vitamin D receptor and a vitamin D activating enzyme, indicating that these cells can produce and respond to activated vitamin D. This suggests that vitamin D deficiency may have a significant impact on inflammatory disorders. We reviewed several recent studies to investigate the mechanisms of vitamin D activity in immune cells and the role of vitamin D in infectious and autoimmune diseases.

## 2. Vitamin D and the Immune Cells

Granulocytes, dendritic cells, monocytes/macrophages, and lymphocytes play an important role in the regulation of the immune system, the inflammatory response, and bone remodeling. In the 1980s, Abe et al. reported that vitamin D induces differentiation of monocytes and macrophages [5]. It has also been demonstrated that dendritic cells, monocytes/macrophages, and T and B cells express vitamin D and 1α-hydroxylase (CYP27B1), the vitamin D-activating enzyme [6]. In this section, we discuss the function of vitamin D in these immune cells, which play important roles in infectious and autoimmune diseases.

### 2.1. Dendritic Cells

A dendritic cell acts as an antigen-presenting cell to T cells, priming the adaptive immune response. Stimulation with the active form of vitamin D downregulates MHC class II and co-stimulatory molecules (such as CD40, CD80, and CD86), expressed on dendritic cells, resulting in T cell activation. In addition, activated vitamin D or vitamin D analogues suppress dendritic cell cytokine production, specifically, interleukin (IL)-12, which affects the differentiation of T helper cells into Th1 cells, and IL-23, which affects the differentiation of T helper cells into Th17 cells. Vitamin D also promotes expression of the anti-inflammatory cytokine IL-10 [7,8].

### 2.2. Monocytes/Macrophages

Monocytes/macrophages play an important role in the protection of infections by producing inflammatory cytokines. Components from bacteria, viruses, and fungi are recognized by toll-like receptors expressed on the surfaces of monocytes and macrophages, which upregulate the expression of the vitamin D receptor (VDR) and CYP27B1 [9]. After transport into the cell, 25-hydroxyvitamin D (25D) is metabolized into the active form of vitamin D, 1,25-dihydroxyvitamin D (1,25D), by CYP27B1. Within the cell, 1,25D binds to VDR, which exists in the cytosol or nucleus, and the activated VDR forms a heterodimer with the retinoid-X receptor (RXR). The heterodimer binds to DNA and induces the production of antibiotic peptides, such as cathelicidin and β-defensin 2 [10,11]. These peptides produce antibiotic effects by destroying the cell membranes of bacteria and viruses or by activating an antibiotic signaling cascade in infected cells.

The NF-κB transcription factor is required for the expression of DEFB4, the gene encoding β-defensin 2, and 1,25D has been shown to affect NF-κB activation. 1,25D induces the expression of nucleotide-binding oligomerization domain 2 (NOD2), an intracellular pathogen-recognizing protein. NOD2 binds to muramyl dipeptide, a common peptidoglycan among gram-negative bacteria, to promote the transcription of DEFB4 via NF-κB [12]. Furthermore, 1,25D has been shown to induce autophagy, a conserved cellular degradation and recycling process in eukaryotes, in macrophages, and to promote antibiotic activity (Figure 1). Yuk et al. reported that 1,25D induces transcription of the autophagy-associated proteins Atg-5 and Beclin-1, which promote autophagy via the induction of cathelicidin and its downstream factors (p38, ERK, and C/EBPβ) [13]. 1,25D produced by monocytes and macrophages induces the expression of cathelicidin and β-defensin 2, which contribute to protection against infections [14]. It also regulates the epigenetic programming of monocytes/macrophages during immune challenges and, thereby, affects immunological memory and subtype differentiation of immune cells [15].

### 2.3. T Cells

T cells interact with antigen-presenting dendritic cells to induce an antigen-specific immune response. T cells express both the VDR and CYP27B1. Naïve T cells express a low level of the VDR, which gradually increases upon activation. Moreover, 1,25D suppresses the proliferation and differentiation of CD4-positive T cells via cytokine secretion. Specifically, 1,25D reduces Th1-type differentiation and the secretion of inflammatory cytokines (IL-2, IFNγ, and TNF-α), and promotes Th2-type differentiation and the secretion of anti-inflammatory cytokines (IL-4, IL-5, and IL-10) [16]. Additionally, 1,25D inhibits the secretion of Th17-related cytokines (IL-17, IFNγ, IL-21, and IL-22) and negatively regulates the RAR-related orphan receptor C and the aryl hydrocarbon receptor, which are master regulators of Th17-type differentiation [17,18]. Conversely, 1,25D promotes the differentiation of regulatory T cells, preventing an increased autoimmune response by inducing the anti-inflammatory cytokine IL-10 and the FoxP3 transcription factor [19].

### 2.4. B Cells

B cells play a key role in autoimmune disease via the production of autoantibodies; they also express the VDR and CYP27B1. Previous studies have indicated that B cell differentiation, proliferation, and antibody production are suppressed by 1,25D-treated T helper cells. However, a recent study demonstrated that 1,25D itself suppresses naïve B cell differentiation or maturation to memory B and plasma cells [20].

## 3. Vitamin D and Infectious Disease

In recent years, epidemiological data have shown that vitamin D deficiency is associated with morbidity in several infectious diseases. However, vitamin D supplementation as a treatment for infectious diseases remains controversial, in part due to conflicting clinical study results [21]. In this section, we will focus on respiratory infections, including the 2019 coronavirus disease (COVID-19) and the flu. We reviewed recent clinical studies on the correlation between vitamin D deficiency and morbidity, the effects of vitamin D supplementation in randomized controlled trials (RCT), and the underlying mechanisms.

### 3.1. Flu and Vitamin D

It has been reported that a low serum level of 25D in patients is positively correlated with morbidity in upper respiratory tract infections, including the flu. The increased morbidity of the flu in the winter may be related to decreased exposure to sunlight since synthesis of the active form of vitamin D requires sunlight. Interestingly, in cases with an increase in the serum 1,25D level by 10 nmol/L, the risk of infection decreases by 7% [22]. However, the use of vitamin D supplementation continues to be controversial. Hayashi et al. found that mice fed a diet consisting of a high dose of 25D and infected with the influenza virus exhibited decreased production of the inflammatory cytokines, IL-5 and IFN-γ [23]. In an RCT conducted by Murdoch, healthy adults were given more than 100,000 IU of vitamin D3 for 1 month; however, morbidity resulting from upper respiratory tract infections did not decrease [24]. Conversely, another RCT conducted by Camargo demonstrated that vitamin D3 administration (300 IU/day for 3 months) resulted in decreased morbidity related to upper respiratory tract infections in Mongolian children [25]. However, another RCT targeting immunodeficient patients in Sweden showed that daily administration of vitamin D3 (4000 IU/day) reduced symptoms, the amount of pathogen detected in mucus, and the duration of antibiotic treatment [26,27]. Moreover, a double-blind trial conducted by Urashima et al. indicated that children treated with vitamin D3 (1200 IU/day) had a significantly lower rate of flu type A (18.6%) compared to the placebo group (10.8%) [28]. In that trial, vitamin D supplementation was significantly effective in children with asthma.

### 3.2. COVID-19 and Vitamin D

COVID-19 is a serious public health threat. Its pathogen, the severe acute respiratory syndrome coronavirus 2 (SARS-CoV-2), causes respiratory symptoms. Many observational studies have demonstrated that serum vitamin D levels are inversely correlated with the incidence and severity of COVID-19 [29]. The suggested mechanism is that vitamin D suppresses the renin-angiotensin system, increases ACE2 concentration in acute lung injury, and induces an interferon (IFN)-mediated antiviral reaction. Xu et al. suggested that 1,25D alleviates lipopolysaccharide-induced acute lung injury through renin suppression and Ang Ⅱ expression [30]. A similar mechanism may be expected in SARS-CoV-2-related acute respiratory distress syndrome (ARDS). Type Ⅰ IFNs are natural antiviral mediators, and there is evidence that their response contributes to COVID-19 severity [31]. A molecular study has described a constitutive inhibitory interaction between unbound VDR and STAT1, a transcription factor in Type Ⅰ IFN signaling. Consequently, vitamin D deficiency could reduce the effectiveness of the IFN-mediated antiviral response due to higher levels of unbound VDR [32]. T Therefore, vitamin D supplementation may contribute to the prevention of severe COVID-19.

## 4. Vitamin D and Autoimmune Disease

Clinical studies have indicated that vitamin D deficiency is positively correlated with the onset, or exacerbation, of various autoimmune diseases [33]. Studies seeking to define the mechanisms underlying this finding are ongoing. However, as in infectious diseases, there is some debate as to whether active vitamin D or vitamin D supplementation improves autoimmune disease pathology. In this section, we review studies on the correlation between serum vitamin D levels and the onset or exacerbation of autoimmune diseases, along with the use of vitamin D in their treatment, and the underlying mechanisms.

### 4.1. RA and Vitamin D

RA is an autoimmune disease that typically involves chronic synovial inflammation and joint destruction, and causes problems with motility. Several studies have reported a correlation between serum 25D levels and RA [34,35]. Caraba et al. reported a significant inverse correlation between 25D levels and disease activity score 28 (DAS28), TNF-α, and IL-6. A significant positive correlation between 25D and endothelial function has also been reported [36]. In a study with 645 early RA patients, vitamin D deficiency correlated with more active and severe disease and was suggested as a useful biomarker to predict disability progression over one year [35]. However, a causal relationship between low 25D levels and disease activity is difficult to establish because it is possible that patients with high disease activity may have reduced exposure to sunlight, which may decrease vitamin D synthesis in the skin [37]. Several studies have investigated the relationship between vitamin D insufficiency/deficiency and the risk of RA, but only a few studies have demonstrated a significant correlation. The role of vitamin D supplementation in the prevention of RA was verified by the Women’s Health Initiative Calcium plus Vitamin D trial. RA morbidity was not lower in the calcium plus vitamin D-treated group compared to the placebo group [38]. Vitamin D has also been used for treatment. A randomized trial conducted in 150 patients from India concluded that weekly supplementation of 60,000 IU in early treatment-naïve RA patients resulted in pain relief [39]. In a meta-analysis of six randomized controlled trials, vitamin D complementary therapy resulted in more beneficial effects on DAS28, ESR. However, improvement was not observed in other parameters such as VAS (Patient Global Pain Score), SJC (Swollen Joint Count), or CRP. Notably, in a subgroup analysis, a significantly improved VAS score was observed with vitamin D supplementation of more than 50,000 IU/week, for more than 12 weeks [40]. The pharmacological mechanism of 1,25D in RA has also been investigated. In vitro, 1,25D at an optimal physiological concentration in combination with corticosteroids additively inhibited the TNF-α, IL-17, IL-6, and matrix metallopeptidase (MMP) production by synoviocytes cocultured with T cells [41]. Furthermore, in VDR-deficient mice, TNF-α-induced arthritis was found to be severe, and more macrophages and fibroblasts were found in the joints [42]. It has been suggested that vitamin D affects the on-site inflammatory cells. Another study reported that 1,25D treatment in adjuvant-induced arthritis (AIA) did not suppress the local production of inflammatory cytokines but suppressed them in the spleen, which suggested that 1,25D has systemic effects [43]. Further studies are needed to reveal the underlying mechanism.

### 4.2. SLE and Vitamin D

SLE is an autoimmune disease characterized by the deposition of immune complexes in tissue and systemic inflammation. Patients with SLE tend to have low levels of 25D; recent studies have indicated that a low level of 25D in patient serum is associated with the SLE disease activity index score [44,45,46,47]. Data from a Chinese study have shown a possible link between vitamin D insufficiency and flare severity [48]. The use of 1,25D has been evaluated in pathological murine models of disease. Lemire et al. reported that the administration of 1,25D to MRL/lpr mice, an SLE mouse model, improved skin lesions [49], but no improvement in renal lesions was observed. However, Deluca et al. found that 1,25D administration did improve renal lesions [50]. One mechanism underlying 1,25D function in SLE has been defined in in vitro studies, as described here. Treatment of monocytes from SLE patients with 1,25D or its analogue resulted in decreased antibody production, particularly antinuclear antibodies produced by the B cells [51]. These results support the suggestion that reducing vitamin D deficiency is an effective method to control SLE disease. However, it is difficult to maintain proper vitamin D levels in SLE patients. One reason is that it is often recommended that SLE patients avoid exposure to the sun because some patients exhibit sensitivity to sunlight. A consensus on the use of vitamin D supplementation in SLE patients has not yet been established [52].

### 4.3. MS and Vitamin D

MS is an autoimmune disease characterized by frequent inflammatory lesions in the central nervous system, which often cause repeated periods of exacerbation and remission. In recent years, the relationship between vitamin D deficiency and MS has become a topic of interest. MS is prevalent in high-latitude regions, possibly because of decreased exposure to sunlight and the consequent reduction in vitamin D synthesis [53]. Serum 25D and 1,25D levels are lower in MS patients compared with healthy volunteers, and serum 25D levels were shown to be associated with disease activity and severity [54]. A case–control study conducted in US soldiers identified 25D deficiency as a risk factor for MS in Caucasians, indicating that vitamin D deficiency is associated with MS pathogenesis [55]. However, it is difficult to establish a causal relationship between vitamin D deficiency and MS. The experimental autoimmune encephalomyelitis (EAE) model can be used to analyze the relationship between vitamin D deficiency and MS. Cantorna et al. revealed that administration of 1,25D prior to inducing EAE prevents disease onset, and treatment with 1,25D after EAE induction inhibits disease progression [56]. One potential mechanism of action may be that 1,25D suppresses IL-12 production in dendritic cells, promoting the production of anti-inflammatory cytokines (such as IL-10), which inhibit the differentiation and proliferation of Th1 cells [7,27,57,58,59] (Figure 2). A recent clinical study conducted in relapsing–remitting MS patients found that a high dose of vitamin D3 (50,000 IU/5 days) taken for 3 months significantly improved mental state compared with the placebo [60]. Further studies on vitamin D treatment of MS are necessary to determine its usefulness.

### 4.4. Autoimmune Endocrine Disorders and Vitamin D

Vitamin D deficiency also causes autoimmune endocrine disorders, including Hashimoto thyroiditis, type-1 diabetes mellitus (T1DM), Addison’s disease, and Graves’ disease [61]. Several studies have suggested that the VDR polymorphism may be associated with autoantibody production and disease susceptibility [62,63]. Furthermore, polymorphism of CYP27B1, another component of vitamin D metabolism, was found to be associated with Addison’s disease [64]. However, the effectiveness of vitamin D supplementation in preventing the progression of these diseases has not been established.

## 5. Conclusions

Vitamin D acts directly on immune cells, which play a key role in autoimmune diseases. Clinical studies have demonstrated that vitamin D deficiency is related to morbidity in infectious diseases and the onset or progression of autoimmune diseases, such as RA, SLE, and MS. Vitamin D supplementation has been utilized to protect against or treat some inflammatory diseases; however, its effectiveness remains unclear. Further study is necessary to determine the mechanisms of activation of vitamin D in each disease and to establish proper treatment strategies for the future.

## Figures and Tables

**Figure 1 biomolecules-11-01624-f001:**
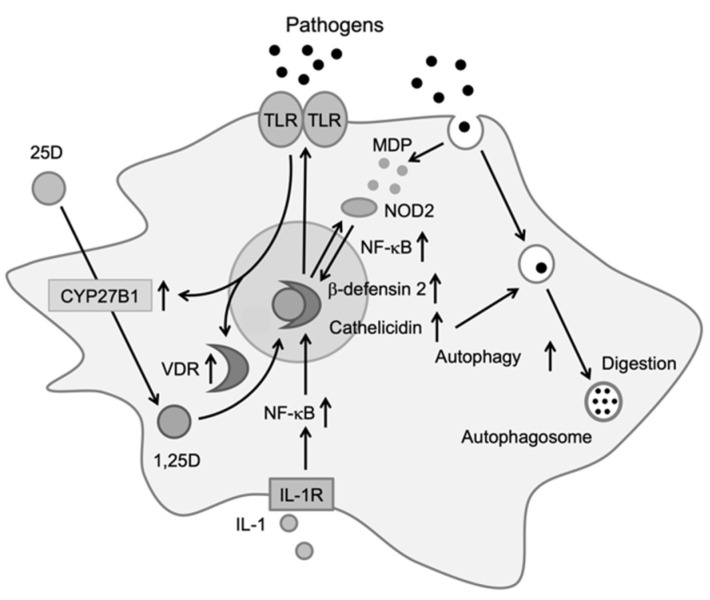
Effects of vitamin D on immune cells. Activation of toll-like receptors by pathogens increases the expression of vitamin D receptor (VDR) and CYP27B1. Upon entering the cell, 25D is metabolized to 1,25D by CYP27B1. 1,25D then binds to VDR, which induces cathelicidin and β-defensin 2. Cathelicidin promotes antibiotic activity via autophagy [14].

**Figure 2 biomolecules-11-01624-f002:**
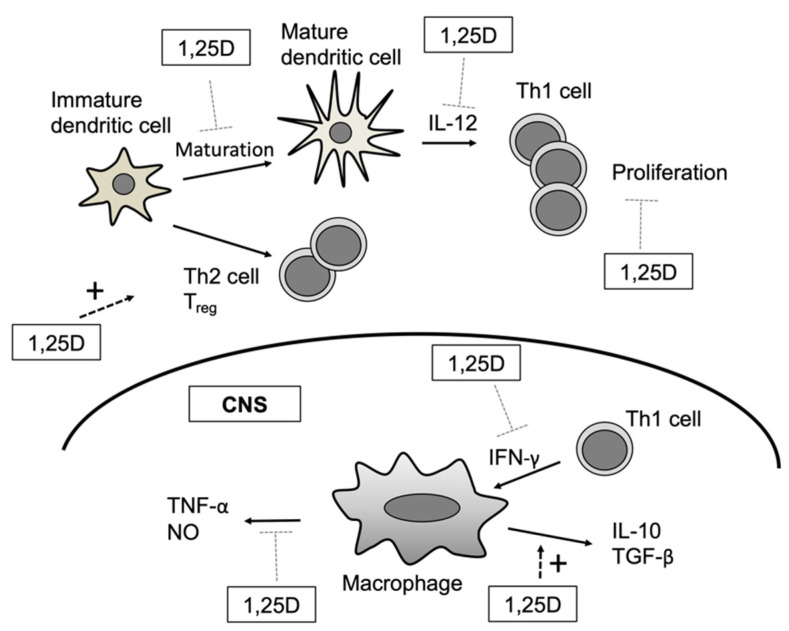
In experimental autoimmune encephalomyelitis (EAE) model, 1,25D inhibits interleukin (IL)-12 production by suppressing maturation of dendritic cells. Consequently, proliferation of Th1 cells is inhibited. On the other hand, 1,25D promotes the differentiation of T cells into Th2 or Treg cells. 1,25D plays an anti-inflammatory role by suppressing proinflammatory cytokines such as TNF-α or by promoting anti-inflammatory cytokines such as IL-10 or TGF-β derived from macrophages [27].

## Data Availability

Not applicable.

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
