# Peer review of "The Effects of Vitamin D on Immune System and Inflammatory Diseases"

_biomolecules, 2021, doi:10.3390/biom11111624_

Round 1

Reviewer 1 Report

The present paper aims to e review the mechanisms of vitamin D activity in immune cells and the role of vitamin D in infectious and autoimmune diseases.

A few changes are needed, as follows:

Introduction: Please mention that vitamin D exerts several pleiotropic effects, related also to cardiovascular disorders (Buleu FN, et al. Correlations between Vascular Stiffness Indicators, OPG, and 25-OH Vitamin D3 Status in Heart Failure Patients. Medicina (Kaunas). 2019 Jun 25;55(6):309. doi: 10.3390/medicina55060309 and Mozos I, et al. Links between Vitamin D Deficiency and Cardiovascular Diseases. Biomed Res Int. 2015;2015:109275. doi: 10.1155/2015/109275) and cancers (Jeon SM, et al.  Exploring vitamin D metabolism and function in cancer. Exp Mol Med. 2018 Apr 16;50(4):1-14. doi: 10.1038/s12276-018-0038-9).

Vitamin D and autoimmune disease: Please mention a few words about vitamin D in Rheumatoid Arthritis Patients (Caraba A, et al. Vitamin D Status, Disease Activity, and Endothelial Dysfunction in Early Rheumatoid Arthritis Patients. Dis Markers. 2017;2017:5241012. doi: 10.1155/2017/5241012).

A link between vitamin D level and autoimmune endocrine disorders should also be mentioned, such as Hashimoto thyroiditis (Sönmezgöz E, et al.  Hipovitaminosis D en niños con tiroiditis de Hashimoto [Hypovitaminosis D in Children with Hashimoto’s Thyroiditis]. Rev Med Chil. 2016 May;144(5):611-6. Spanish. doi: 10.4067/S0034-98872016000500009), type 1 diabetes mellitus, Addison disease, Graves disease (Altieri B, et al. Does vitamin D play a role in autoimmune endocrine disorders? A proof of concept. Rev Endocr Metab Disord. 2017 Sep;18(3):335-346. doi: 10.1007/s11154-016-9405-9).

Author Response

Response to Reviewer #1:

[Comment]

Introduction: Please mention that vitamin D exerts several pleiotropic effects, related also to cardiovascular disorders (Buleu FN, et al. Correlations between Vascular Stiffness Indicators, OPG, and 25-OH Vitamin D3 Status in Heart Failure Patients. Medicina (Kaunas). 2019 Jun 25;55(6):309. doi: 10.3390/medicina55060309 and Mozos I, et al. Links between Vitamin D Deficiency and Cardiovascular Diseases. Biomed Res Int. 2015;2015:109275. doi: 10.1155/2015/109275) and cancers (Jeon SM, et al.  Exploring vitamin D metabolism and function in cancer. Exp Mol Med. 2018 Apr 16;50(4):1-14. doi: 10.1038/s12276-018-0038-9).

[Response]

   We agree with this point, and have mentioned cardiovascular disorders and cancers in the revised manuscript as follows: “Due to the pleiotropic effects of vitamin D, its deficiency is related to a higher risk of cardiovascular diseases [1,2], infectious diseases, and autoinflammatory diseases, such as rheumatoid arthritis (RA), systemic lupus erythematosus (SLE), and multiple sclerosis (MS).”. (Page 1, line 25–page 1, line 28) “Anti-cancer effects of vitamin D have found application in cancer treatment”. (Page 1, line 30 – page 1, line 31)

[Comment]

Vitamin D and autoimmune disease: Please mention a few words about vitamin D in Rheumatoid Arthritis Patients (Caraba A, et al. Vitamin D Status, Disease Activity, and Endothelial Dysfunction in Early Rheumatoid Arthritis Patients. Dis Markers. 2017;2017:5241012. doi: 10.1155/2017/5241012).

[Response]

We agree that rheumatoid arthritis is the prototypical autoimmune disease. We have created a new subsection in Section 4: Vitamin D and autoimmune disease, entitled “RA and vitamin D”. We have mentioned the correlations between 25D level and disease activity, and rheumatoid arthritis prevention and treatment using vitamin D, along with the involved mechanisms, in this subsection. (Page 5, line 5–page 5, line 28)

[Comment]

A link between vitamin D level and autoimmune endocrine disorders should also be mentioned, such as Hashimoto thyroiditis (Sönmezgöz E, et al.  Hipovitaminosis D en niños con tiroiditis de Hashimoto [Hypovitaminosis D in Children with Hashimoto’s Thyroiditis]. Rev Med Chil. 2016 May;144(5):611-6. Spanish. doi: 10.4067/S0034-98872016000500009), type 1 diabetes mellitus, Addison disease, Graves disease (Altieri B, et al. Does vitamin D play a role in autoimmune endocrine disorders? A proof of concept. Rev Endocr Metab Disord. 2017 Sep;18(3):335-346. doi: 10.1007/s11154-016-9405-9).

[Response]

We have created a new subsection in Section 4: Vitamin D and autoimmune disease, entitled “Autoimmune endocrine disorders and vitamin D”.

(page 6, line 17-page 6,line 24)

Reviewer 2 Report

In this review, the authors describe the role of the vitamin D on the different members of the immune system.

Major concerns:

  1. This review does not bring anything new. The newest citation is from 2016, and from that year are only two citations, one from 2015, from 2014, and from 2013, although they are writing about “recent studies”. Since then many other reviews on this subject have been written, which aren’t mentioned. All the new insights on the role of vitamin D on viral infection is also missing. Only in 2021 there are more than 300 hits in PubMed, if searching for “vitamin D and inflammatory diseases”.
  2. There are also some factual mistakes: in chapter 2.2. the authors write “the activated VDR forms heterodimers, which induce the production of antibiotic peptides”. It was not mentioned with which receptor form the VDR the heterodimers. Then the heterodimers cannot induce the production of anything per se, they have to bind to the DNA and induce transcription. They don’t even mention that the VDR is a transcription factor or nuclear receptor.

Minor issue:

The ref. 8 and 9 are mixed up, first should have been 9 cited (Wang et al), then 8 (Yuk et al.).

Author Response

Response to Reviewer #2:

[Comment]

This review does not bring anything new. The newest citation is from 2016, and from that year are only two citations, one from 2015, from 2014, and from 2013, although they are writing about “recent studies”. Since then many other reviews on this subject have been written, which aren’t mentioned. All the new insights on the role of vitamin D on viral infection is also missing. Only in 2021 there are more than 300 hits in PubMed, if searching for “vitamin D and inflammatory diseases”.

[Response]

We have added recent citations, from 2016 onwards, especially in Section 3. We have also included the relationship between COVID-19 and vitamin D in this section, Vitamin D and infectious disease (page 4, line 10–page 4, line 25). We have also excluded the word “recent” from the descriptions of studies before 2017.

[Comment]

There are also some factual mistakes: in chapter 2.2. the authors write “the activated VDR forms heterodimers, which induce the production of antibiotic peptides”. It was not mentioned with which receptor form the VDR the heterodimers. Then the heterodimers cannot induce the production of anything per se, they have to bind to the DNA and induce transcription. They don’t even mention that the VDR is a transcription factor or nuclear receptor.

[Response]

Thank you for pointing this out. We have added the name of the receptor that forms the VDR heterodimer. We have further explained that VDR exists in cytosol or nucleus without vitamin D, and that activated VDR binds to DNA. The manuscript has been revised as follows: “Within the cell, 1,25D binds to VDR, which exists in cytosol or nucleus, and the activated VDR forms a heterodimer with the retinoid-X receptor (RXR). The heterodimer binds to DNA and induces the production of antibiotic peptides, such as cathelicidin and β-defensin 2.” (page 2, line 18-page 2,line 23).

[Comment]

The ref. 8 and 9 are mixed up, first should have been 9 cited (Wang et al), then 8 (Yuk et al.).

[Response]

Thank you for pointing this out. We have corrected the order of citations.

Reviewer 3 Report

Finding new therapeutic options for autoimmune diseases is a challenge for scientists. The article „The effects of vitamin D on immune system and inflammatory diseases” is an in interesting article, but it should be revised before publication.

First of all the authors should mention why they decided to choose only 2 autoimmune diseases: lupus and multiple sclerosis.

The aim of this review should be clearly written.

The authors should also mention what was the methodology, how they select the articles.

Also, I recommend to insert a part in this review where the authors explain the connection between immune cells influenced by vitamin D and the pathogenesis of described autoimmune diseases. It seems that the review has 2 parts without connection between them.

Author Response

Response to Reviewer #3:

[Comment]

First of all, the authors should mention why they decided to choose only 2 autoimmune diseases: lupus and multiple sclerosis.

[Response]

Thank you for the comment. We have added sections describing rheumatoid arthritis, the representative autoinflammatory disorder, and autoinflammatory endocrine disorders.

[Comment]

The aim of this review should be clearly written.

[Response]

     The aim of this review was stated at the end of the Introduction. We have expanded on it and revised it as follows.

“This suggests that vitamin D deficiency may have a significant impact on inflammatory disorders. We reviewed several recent studies to investigate the mechanisms of vitamin D activity in immune cells and the role of vitamin D in infectious and autoimmune diseases.” (page 1, line 34-page 1,line 37).

[Comment]

The authors should also mention what was the methodology, how they select the articles.

[Response]

    Thank you for the comment. We have added a summary of the methodology at the beginning of each section. Briefly, we reviewed the studies that investigated the correlation between vitamin-D deficiency and morbidity, the effects of vitamin-D supplementation, and the underlying mechanisms.

[Comment]

Also, I recommend to insert a part in this review where the authors explain the connection between immune cells influenced by vitamin D and the pathogenesis of described autoimmune diseases. It seems that the review has 2 parts without connection between them.

[Response]

    Thank you for the suggestion. Pathological models have hypothesized various effects of vitamin D on immune cells, but not all the in vitro effects have been confirmed in vivo. We have mentioned the effects of vitamin D on immune cells in the subsections on rheumatoid arthritis (page5, line21-page 5, line27) and multiple sclerosis (page 6, line 9–page 6, line 12).

Round 2

Reviewer 2 Report

The review is OK, but in my opinion this review doesn’t bring much new information that would vouch for its publication in such a reputable journal. I would have expected that the new papers in this field (more around 1000) in the last 3 years would be reviewed. The reference was a tiny bit updated: only one paper from 2021, 2020, 2019, and 2018 each is really meagre. 4 citations from 2017 are a small improvement.

Author Response

Response to Reviewer 2:

[Comment]

The review is OK, but in my opinion this review doesn’t bring much new information that would vouch for its publication in such a reputable journal. I would have expected that the new papers in this field (more around 1000) in the last 3 years would be reviewed. The reference was a tiny bit updated: only one paper from 2021, 2020, 2019, and 2018 each is really meagre. 4 citations from 2017 are a small improvement.

[Response]

   We added another 14 citations from 2018, 2019, 2020, 2021 to several sections (References: [3][15][16][21][23][33][34][35][39][40][41][46][47][48]. We hope that the revised manuscript is suitable for publication.

Reviewer 3 Report

The article could be published in the present form

Author Response

Response to Reviewer :

Thank you very much for kindly giving us a comment.